# A Benchmark for Scalable Oversight Mechanisms

## Abstract

As AI agents surpass human capabilities, *scalable oversight* – the problem of effectively supplying human feedback to potentially superhuman AI models – becomes increasingly critical to ensure alignment. While numerous scalable oversight mechanisms have been proposed, they lack a systematic empirical framework to evaluate and compare them. While recent works have tried to empirically study scalable oversight mechanisms – particularly Debate – we argue that they contain methodological flaws that limit their usefulness to AI alignment. We introduce the *scalable oversight benchmark*, a principled framework for evaluating human feedback mechanisms based on our agent score difference (ASD) metric, a measure of how effectively a mechanism advantages truth-telling over deception. We supply a Python package to facilitate rapid and competitive evaluation of scalable oversight protocols on our benchmark, and conduct a demonstrative experiment benchmarking Debate.

## 1 Introduction

One way to frame the limitations of currently widely-used alignment techniques such as reinforcement learning from human feedback (Christiano et al., 2017), is that they fundamentally rely on a human's ability to judge the correctness or value of a (potentially superhuman) AI's outputs (Burns et al., 2024). In other words, the AI model is trained on the human supervisor's *immediate, superficial* volition, rather than on her *extrapolated volition* (Yudkowsky, 2004).

The problem of developing a human feedback mechanism that scales to superhuman intelligences is known as *scalable oversight* (Bowman et al., 2022). Broadly speaking, there are two ways to think about the scalable oversight problem:

1. The problem of developing a **training method** that makes honesty (or more generally "alignment") the best policy for the model; i.e. something to replace or extend RLHF to the superhuman realm.

2. An **inference-time oversight mechanism** to catch a model when it says something false or does something bad; i.e. a mechanism design problem to get AIs to be truthful or useful.

For example, in *Debate*, the most widely-known scalable oversight protocol introduced in Irving et al. (2018), the model is incentivized to tell the truth if it knows that a lie can be caught and convincingly refuted by its opponent. A list of other competing proposals is given in Section 1.1.

While there is much mathematical and intuitive elegance underlying each of these mechanisms, the diversity of these proposals and their theoretical claims to superiority begs the question: ***how can we evaluate and compare scalable oversight protocols themselves?***

One approach, taken by recent works such as Radhakrishnan (2023); Michael et al. (2023); Khan et al. (2024); Kenton et al. (2024); Arnesen et al. (2024)[1], is to evaluate these protocols

---

[1] we will collectively refer to these papers as "previous debate experiments" or "Previous work" when making comments that apply to all of them

(specifically Debate) *empirically*, by measuring their effect on the accuracy of the "judge" (the human or weak model providing feedback).

Building and improving on their work, we introduce the **scalable oversight benchmark**, a *principled and general empirical framework* for evaluating human feedback mechanisms for their impact on AI alignment. Specifically, our contributions in this work are as follows.

**1) A principled metric for evaluating scalable oversight protocols.** In previous debate experiments, protocols were evaluated based on "judge accuracy" – i.e. they looked at how much Debate improved a (human or weak model) judge's accuracy at answering questions relative to a baseline "Consultancy" protocol. In Section 2 we argue that this is the wrong metric to evaluate scalable oversight protocols on from an alignment or safety point-of-view. Instead we introduce the **agent score difference** *metric*, which measures how much the protocol "advantages truth over falsehood", i.e. the difference in the score earned by an agent arguing for the true answer vs. for the false answer. For example, if under some scalable oversight protocol, a judge believes a truthful agent with probability 0.8 and a lying agent with probability 0.6, the ASD is $\log(0.8) - \log(0.6) \approx 0.29$. This measure is equivalent to judge accuracy for *Simultaneous Debate*; however it is not equivalent for Consultancy, hence the baseline comparison in previous debate experiments is incorrect.

**2) A library for conducting systematic evaluations on scalable oversight protocols.** We characterize the class of experiments done to evaluate Debate in Previous work and generalize it to any scalable oversight protocol – and further provide a Python library `SOlib`[2] to enable performing *principled* and *systematic* experiments evaluating scalable oversight protocols on our metric and meaningfully comparing between them. One may use our package by simply subclassing our `Protocol` class and running its `experiment` method on any choice of agent and judge models and a labelled dataset of questions.

**3) Experiments with tool use.** Scalable oversight is desired for settings with a significant *capabilities asymmetry* between the agent (e.g. debater) and the judge, as it is intended to be used for judging superhuman AI models. Previous debate experiments has implemented this mainly by simulating this capabilities asymmetry with information asymmetry (Radhakrishnan, 2023; Khan et al., 2024), and by using larger and more capable models for the agent than for the judge or allowing chain-of-thought tokens for the agent (Kenton et al., 2024). We introduce a third dimension to asymmetry: *tool use.* Specifically, we run our benchmark for the *Debate* and *Consultancy* protocols on a demonstrative sample of the GSM8K dataset[3], with only the agent (but not judge) equipped with a simple calculator tool. The experiment is currently running (the full configuration is given in Appendix A) and not yet complete; its results will be reported in a full paper after the workshop, or in the final version of the paper if permitted.

Our vision is a world where alignment researchers can rapidly prototype scalable oversight protocols and evaluate them on our benchmark, creating competitive pressures for better mechanisms. The

## 1.1 Related work

**Scalable Oversight.** Apart from Debate, proposed mechanisms for scalable oversight include: *Iterated Amplification* (Christiano et al., 2018), *market-making* (Hubinger, 2020), *self-critique* (Saunders et al., 2022), *reward-modelling* (Leike et al., 2018) and proposed improvements to Debate such as *doubly-efficient debate* (Brown-Cohen et al., 2024). A slightly dated review and discussion of these can be found in Bowman et al. (2022).

**Weak-to-strong generalization and human feedback.** Scalable oversight can be seen as an approach to *weak-to-strong generalization* (Sang et al., 2024; Lang et al., 2025) that explicitly relies on the weak model (or human) providing reward to a strong model (as

---

[2] https://anonymous.4open.science/r/math_problems_debate-F4B4

[3] Cobbe et al. (2021), a dataset of grade-school math word problems

opposed to e.g. fine-tuning or transfer learning). The relationship between scalable oversight and human feedback mechanisms is made explicit by e.g. Cheng et al. (2024), who consider *reinforcement learning from debate feedback*.

**Previous Debate Experiments.** The following works: Radhakrishnan (2023); Michael et al. (2023); Khan et al. (2024); Kenton et al. (2024); Arnesen et al. (2024), all of which apply a similar empirical lens to the scalable oversight problem (specifically Debate), represent the closest line of work to ours. While there are important differences and subtleties in the experiments they conduct, their methodology can in broad terms be described as measuring the effect of Debate on *judge accuracy*, relative to a baseline of "Consultancy" (a mechanism where the AI is randomly assigned one of the two answers to argue for, and does not have an adversary).

## 2  THE AGENT SCORE DIFFERENCE METRIC

Consider a binary question $Q$ with possible answers $A \in \{0, 1\}$ (i.e. denoting "NO" and "YES"). The "judge" (human or weak model) is asked to give probabilities $p_1$, $p_0$ for these answers after witnessing the outputs (under some scalable oversight protocol, e.g. Debate) of some stronger "agent".

The straightforward and naive way to evaluate a scalable oversight protocol, taken in previous debate experiments, is to measure "judge score" (e.g. $\log(p_\top)$ where $\top$ is the correct answer, or just $\{p_\top > 0.5\}$ for plain "judge accuracy") – the more accurate the judge's beliefs are under the protocol, the better it is. However, some simple examples demonstrate the weakness of this measure.

**Example 2.1** (`OpenTrust`)**.** Consider a protocol `OpenTrust`, where the AI is allowed to freely choose its answer $A$, and the judge always believes it, i.e. reports $p_A = 1$, $p_{\neg A} = 0$. But since current LLMs are already quite truthful, this would beat any scalable oversight protocol with weak judges and strong agents[4] – even though this is equivalent to having *no* scalable oversight at all!

**Example 2.2** (The weak baseline problem)**.** The most natural baseline to evaluate scalable oversight protocols against would be `NaiveJudge`, where the judge reports its probability without any consultation with a stronger model. But this is a weak baseline: for example, if $Q$ a comprehension task and we place an information asymmetry between the debater and the judge, then the judge sees *less information* in the `NaiveJudge` case than in the `Debate` case. While this problem is clearest in the information asymmetry case, it is true much more generally: when comparing scalable oversight protocols, it is not obvious how to adjust for differences in the amount of "information" or "intelligence" delivered to the judge.

To mitigate the weak baseline problem, previous debate experiments instead used *Consultancy* (or more accurately "Random Consultancy" – where the judge only listens to one side, the Consultant, who has a 50% chance of arguing for the right or wrong answer) as a baseline. However as Kenton et al. (2024) notes, this is also a weak baseline: the result "Debate leads to better judge accuracy than Random Consultancy" is conceptually equivalent to "always hearing arguments for both sides is better than hearing only the wrong side 50% of the time". While this is a worthwhile empirical result, it is only a "sufficing" result, and does not help the goal of comparing across protocols or competitively optimizing for the "best" protocol.

Instead we propose: the *agent score difference* metric: the relative reward received by the agent if it argues for the true answer as opposed to the false answer[5]:

$$ASD = \log p_\top - \log p_\perp$$

---

[4]straightforward experiments with a sample of the GSM-8K dataset demonstrated that, as expected, `claude-3-5-sonnet-20241022` chose to argue for the correct answer over 90% of the time, and 100% of the time when explicitly told what the correct answer is

[5]for elegance, we use log scores in showing math; in practice it is better to use Brier scores to avoid infinities.

Here: $p_\top$ and $p_\perp$ are the judge probabilities for the true and false answer *after listening to an AI arguing for that answer under the scalable oversight protocol*. Importantly: it is *not* true in general that $p_\top = 1 - p_\perp$, because these represent the judge's probabilities in two different worlds: the world in which the agent argues for the true answer, and the world in which the agent argues for the false answer. Only in *simultaneous*[6] *debate with identical debaters* are these worlds identical (because debate is "symmetric" – the protocol creates an opposite debater for any position), and ASD becomes equivalent to judge score. This is not true for e.g. Consultancy – to evaluate Consultancy as a scalable oversight protocol, you need to make *two different runs* for the truthful and untruthful consultant.

One may revisit the examples from earlier and observe that `NaiveJudge` can now be used as a baseline, and that both it and `OpenTrust` would have zero ASD.

## 2.1 Expected Agent Score

Calculating ASD for a scalable oversight mechanism with currently existing models gives us an estimate of how effective that mechanism is *now*, i.e. how good of an alignment training signal they provide to current-day models. However, we would like to know if they will continue to provide a useful training signal as models' capabilities improve.

Previous debate experiments used debaters' ELO ratings as a measure of their capabilities (or "persuasiveness" as it is called in Khan et al. (2024)), and correlated them with judge accuracy. We generalize this to other scalable oversight mechanisms by defining "Expected Agent Score" as a pure measure of the agent's capabilities:

$$EAS = (\log p_\top, \log p_\perp) \cdot \mathbf{p}$$

Where $\mathbf{p}$ is the "propensity to argue for each answer". If we take $\mathbf{p} = (1/2, 1/2)$ we recover random assignment of answers as in previous debate experiments. More realistically we may take:

$$\mathbf{p} = \left( \frac{e^{ASD/\beta}}{1 + e^{ASD/\beta}}, \frac{1}{1 + e^{ASD/\beta}} \right)$$

Where $\beta$ is some temperature parameter (e.g. $\beta = 0$ implies the AI always just picks the answer with higher score whereas $\beta = \infty$ implies the AI randomly chooses).

By correlating ASD with EAS, we can extrapolate how ASD will change with increasing model capabilities under a given scalable oversight mechanism.

## 3 Experimental framework

We now describe precisely our experimental framework for estimating agent score difference – equivalently, we describe the `Protocol` class, in pseudo-Python.

We have a class `Agent` with a method `__call__(context, answer_case)`, which ***simulates*** what an agent giving a particular answer to a question *would* say long-form. This `answer_case` is really a stand-in for the general "alignment" of the AI – whether it argues for the true answer or the false answer, or for a valuable answer or a less valuable answer, etc. With instruction-following language models like we have now, we can simulate different alignments through prompting, as long as we know ground truth.

```python
class Agent:
  def __call__(context, answer_case) -> str:
    ...
```

Then the class `Protocol` determines what reward an agent would get for having a particular alignment:

---

[6]i.e. where both debaters make their arguments at the same time, rather than turn-by-turn

```
class Protocol:

  def __init__(self, judge, ...):
    ...

  @abstractmethod
  def run(self, agent, question, answer_case, ...) -> Prob:
    # This should be subclassed. E.g. for Simultaneous Debate:
    context = [question]
    adversary_answer = not answer_case # 'B' if answer_case == 'A'
                                       #     else 'A'
    for i in num_turns:
      agent_response = agent(context, answer_case)
      adversary_response = adversary(context, adversary_answer)
      context.append(agent_response)
      context.append(agent_response)
    return self.judge(context, answer_case)

  def agent_score_difference(self, agent, question):
    probs = {
      answer_case: self.run(agent, question, answer_case)
      for answer_case in question.answer_cases
    }
    return dot(log(probs), question.answer_values) # e.g. {'A': -1, 'B
                                                   #     ': 1}
```

## 3.1 Limitations of our framework

The key limitation of our experimental framework (which includes that of previous debate experiments) is that it only computes the reward given to *simulated* aligned or mis-aligned behaviour – i.e. it only works because existing language models follow instructions and act truthfully or untruthfully when we tell them to. If you tested with models that are not good at following instructions (e.g. consistently refuse instructions to lie), then your results will mean nothing, because the "agent arguing for a lie" is not actually a faithful simulation of a misaligned AI.

Secondly, our experiments can only be conducted with currently-existing AIs. As discussed in Section 2.1, we may observe and extrapolate trends in ASD vs EAS to predict ASD in the superhuman regime – however, this is not a formal guarantee. Similarly, it is possible that mechanisms that do well on our experiments only work so well on domains where ground truth is available.

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

## A   EXPERIMENTAL SET-UP

- **protocols:**
  - `NaiveJudge`
  - `Propaganda`, where the judge hears arguments from exactly one side[7]
  - `Consultancy`, where the judge hears arguments from exactly one side but may ask follow-up questions interactively, with the following settings: `consultant_goes_first = True, False` and `num_rounds = 2, 4`
  - `Debate` with the following settings: `simultaneous = True, False` and `num_rounds = 2, 4`.
- **questions:** 100 questions from the GSM8K dataset (Cobbe et al., 2021), with plausible wrong answers generated synthetically
- **agents:** (i.e. models for debaters, consultants) a cartesian product of:
  - **models:** `claude-3-5-sonnet-20241022`, `claude-3-5-haiku-20241022`, `claude-3-opus-20240229`, `deepseek-v3`
  - **tools:** `None, [calculator]`
  - **best-of-N:** 1, 4
- **judges:** raw `gpt-4o-mini`, `ollama_chat/llama3.1:8b-instruct-q6_K` with basic prompting

---

[7]recall, once again, that to compute our metrics we do two separate runs where it hears arguments from two separate sides and compute the difference in agent score between these worlds

