# OpenReview forum: "A Benchmark for Scalable Oversight Mechanisms"
_ICLR.cc/2025/Workshop/BuildingTrust — Submitted to BuildingTrust_

### Official Review · Reviewer_G33D · 2025-03-01
**This paper presents a benchmark for evaluating scalable oversight mechanisms, introducing the Agent Score Difference (ASD) metric as a more robust alternative to judge accuracy. The authors provide a Python library (SOlib) for systematic experimentation and demonstrate its use in evaluating the Debate protocol.**

**Rating:** 6
**Confidence:** 3

**Review:**

## Strengths

* The paper introduces the Agent Score Difference (ASD) metric, which directly measures how much a scalable oversight mechanism favors truthfulness over deception. ASD is calculated with the given formula: $ASD = \log p_{\top} - \log p_{\bot}$. This is a significant improvement over previous methodologies that relied solely on judge accuracy.

* The inclusion of the SOlib Python package enables researchers to systematically evaluate scalable oversight protocols, lowering the barrier for further experimentation in AI alignment.

* The paper models real-world scenarios where advanced AI systems may access external tools to improve their responses by including AI tool use in the evaluation framework.

## Weaknesses

* The Related Works section could be more descriptive of other proposed mechanisms for scalable oversight. Currently, it is vague and only glosses over prior research. Since the paper frequently critiques discrepancies in other methodologies, a more thorough discussion of alternative approaches would strengthen its comparative analysis.

* The paper lacks a theoretical proof that high ASD values correlate with robust long-term AI alignment, particularly in adversarial or deceptive AI settings.

* The benchmark is tested only in controlled AI simulation environments and does not incorporate human-AI interaction studies

* The paper critiques prior work for using Consultancy as a weak baseline, yet it still compares its ASD metric to judge accuracy in certain cases, which may introduce similar issues in evaluating effectiveness.

---

### Official Review · Reviewer_eVhR · 2025-03-02
**Incomplete Work and Unclear Contributions**

**Rating:** 3
**Confidence:** 4

**Review:**

This paper provides a framework to compare scalable oversight mechanisms. This is an important problem as we observe the increasing capabilities of large language models from tool use and increased scaling. First, the paper introduces an "agent score difference" metric for evaluating scalable oversight mechanisms. They argue that existing baselines of random consultancy are conceptually flawed and provide arguments in the case where the judge always believes the agent. Moreover, they argue for a more appropriate mechanism in measuring how much the agent is incentivized to provide the true answer. Their proposed arguments could be written with additional clarity and potentially would benefit from theoretical formalization. As it stands, the arguments appear ad-hoc and intuitive, and the authors do little to make any formal claims that their claimed metric would lead to superior alignment. They then introduce a library to enable the testing of their method. However, though they claim to propose an additional benchmark, their implementation details are minimal and superficial -- describing only a class interface without situating it in any real scalable oversight task. Moreover, though they claim this to be a benchmark, there are no concrete experiments conducted in the work and they provide no justifications for their claims besides intuitive argument. They speculatively promise experiments in a future or camera ready version of the paper but it is not possible to evaluate this paper without any experimental results. Besides, the manuscript has incomplete sentences and a minimal appendix. For this reason, I believe it would be most appropriate for this paper to be submitted at a time where the authors can provide more extensive and concrete results -- especially given that this paper purports to introduce a baseline. It is not currently in the shape to be evaluated under peer review.

---

### Decision · Program_Chairs · 2025-03-04

**Decision:**

Reject

**Comment:**

The paper lacks theoretical formalization to support its proposed Agent Score Difference (ASD) metric, making its arguments appear ad-hoc and intuitive rather than rigorously justified. Additionally, it fails to present concrete experiments or implementation details beyond a superficial class interface, making it difficult to evaluate its claims, especially given that it purports to introduce a new benchmark without empirical validation.